# 3D cell aggregates amplify diffusion signals

**Hamidreza Arjmandi[1], Kajsa P. Kanebratt[2], Liisa Vilén[2], Peter Gennemark[2,3], Adam Noel[4]** *

**1** Department of Cancer and Genomic Sciences, College of Medicine and Health, University of Birmingham, Birmingham, United Kingdom, **2** Drug Metabolism and Pharmacokinetics, Research and Early Development, Cardiovascular, Renal and Metabolism (CVRM), AstraZeneca, Gothenburg, Sweden, **3** Department of Biomedical Engineering, Linköping University, Linköping, Sweden, **4** School of Engineering, University of Warwick, Coventry, United Kingdom

* adam.noel@warwick.ac.uk

**Data Availability Statement:** All relevant data are within the manuscript and its Supporting information files.

**Funding:** The authors acknowledge funding from the UK's Engineering and Physical Sciences

## Abstract

Biophysical models can predict the behavior of cell cultures including 3D cell aggregates (3DCAs), thereby reducing the need for costly and time-consuming experiments. Specifically, mass transfer models enable studying the transport of nutrients, oxygen, signaling molecules, and drugs in 3DCA. These models require the defining of boundary conditions (BC) between the 3DCA and surrounding medium. However, accurately modeling the BC that relates the inner and outer boundary concentrations at the border between the 3DCA and the medium remains a challenge that this paper addresses using both theoretical and experimental methods. The provided biophysical analysis indicates that the concentration of molecules inside boundary is higher than that at the outer boundary, revealing an amplification factor that is confirmed by a particle-based simulator (PBS). Due to the amplification factor, the PBS confirms that when a 3DCA with a low concentration of target molecules is introduced to a culture medium with a higher concentration, the molecule concentration in the medium rapidly decreases. The theoretical model and PBS simulations were used to design a pilot experiment with liver spheroids as the 3DCA and glucose as the target molecule. Experimental results agree with the proposed theory and derived properties.

## Introduction

The term *3D cell aggregate (3DCA)* refers to any type of *in vitro* model in which cells are grown in three dimensions, as opposed to the traditional 2D monolayer culture. Organoids, spheroids, and tumoroids are all examples of 3DCA models, each with unique characteristics and applications. 3DCAs have received great attention and popularity in recent years due to their ability to better mimic the complex microenvironments and cell-cell interactions within *in vivo* tissues, in comparison to traditional 2D cell cultures [1]. 3DCAs can be composed of several different cell types, e.g., hepatocytes and stellate cells to form liver spheroids [2]. They can be used to model organ development and disease progression and they have a wide range of applications in basic biological research, drug discovery, and regenerative medicine [3]. Organ-on-chips (OoCs) as bioengineered microdevices use the 3D nature and arrangement of 3DCAs to recapitulate key functional properties of organs and tissues [4].

Research Council, grant number EP/V030493/1.
AstraZeneca provided support in the form of
salaries for K.PK., L.V., P.G. The specific roles of
these authors are articulated in the 'author
contributions' section. The funders had no role in
study design, data collection and analysis, decision
to publish, or preparation of the manuscript.

Mass transfer models are a crucial category of biophysical models that can enable researchers to study the transport of essential molecules, including nutrients, oxygen, signaling molecules, and drugs in 3DCAs. Specifically, the transfer of diffusible nutrients, like oxygen, plays a vital role in regulating fundamental cellular processes such as cell migration, death, and progression through the cell cycle [5, 6]. 3DCAs rely on a culture medium that fills the extracellular space within the cell aggregate and creates a continuous fluidic environment where molecules are transferred through two transport mechanisms, diffusion and flow, which can both contribute to the mass transfer model. When there is slow flow, or in avascular 3DCAs where there is no vascular system to deliver nutrients and oxygen, the diffusion mechanism dominates in the mass transfer model. Diffusion of molecules in an environment is described by Fick's laws of diffusion which are defined by partial differential equations that describe the change in concentration of diffusing molecules with respect to time and space [7].

The diffusion of molecules within the medium outside and inside a 3DCA should be modeled differently to account for the varying physical and chemical properties of the porous structure of the 3DCA. A simplified model for medium diffusion inside a 3DCA is to treat the 3DCA as a porous medium with a corresponding diffusion coefficient that scales down the diffusion coefficient in the free culture medium. To model diffusion of molecules within the medium inside and outside the 3DCA, one can consider two diffusion environments with different diffusion coefficients. The concentration of diffusing molecules is characterized through two partial differential equations. These two equations are connected by defining two boundary conditions (BCs) at the 3DCA border that relate the concentration of molecules inside and outside of the boundary. The first BC is the flow continuity condition, which is applied at the border to ensure equal mass flux across the boundary. The second BC is characterized by the concentration ratio inside and outside of the 3DCA border [7]. This BC, referred to as the *amplification BC* throughout the rest of this paper, accounts for the influence of the 3DCA on the diffusive transport of molecules and has a significant impact on the accuracy of the diffusion model.

Despite its importance, the amplification BC has not been comprehensively characterized in the literature, and different types of models have been proposed by various authors. Some works define the amplification BC as a unitary ratio, implying that the concentration is the same inside and outside the 3DCA border [8–11]. Astrauskas et al. (2019) presented a reaction-diffusion equation model for the analysis of dye penetration into cellular spheroids in which the unitary ratio is employed to model the boundary condition [8]. Leedale et al. (2020) proposed a diffusion model for drug transport and metabolism that is developed in a multiscale spheroid framework, accounting for microscale processes where the concentration at the boundary is simply assumed to be equal to the outside concentration [9]. Bull et al. (2021) developed a hybrid, off-lattice agent-based model for oxygen-limited spheroid growth [10]. The concentration at the spheroid boundary is assumed to be maintained at the constant oxygen concentration in the culture medium, i.e., the unitary ratio is implied. As part of his thesis, Rousset (2022) studied the molecule diffusion into and consumption by a spheroid where the unitary ratio is used to model the boundary condition [11].

In other works, a porosity ratio is assumed for the 3DCA BC, where the concentration inside the boundary is expressed as the concentration outside the boundary multiplied by the porosity coefficient. Graff et al. (2003) developed a mathematical model to provide an improved understanding of the quantitative interplay among the rate processes of diffusion, binding, degradation, and plasma clearance in antibody penetration of tumor spheroids [12]. The authors considered the concentration at the spheroid boundary to be equal to the concentration outside multiplied by the spheroid porosity. In other words, they assumed that the molecule concentration at the *extracellular space* boundary is equal to the concentration outside

the spheroid. A similar boundary condition has been considered by Goodman et al. (2008), where the authors developed a mathematical model of nanoparticle penetration into multicellular spheroids that accounts for radially-dependent changes in tumor architecture [13].

Furthermore, some authors have modeled the diffusion process only inside the 3DCA where they consider a constant concentration inside the boundary but do not relate it to the concentration outside the boundary [14, 15]. Grimes et al. (2014) employed an oxygen diffusion model in three-dimensional tumor spheroids to present a method for estimating rates of oxygen consumption from spheroids [14]. The concentration at the boundary of the tumor is assumed to be constant and unrelated to the concentration in the surrounding media. In other words, a model with one diffusion environment is assumed where the concentration inside the border is fixed. Klowss et al. (2022) developed a stochastic model that provides quantitative information about nutrient availability within a spheroid [15]. Similar to Grimes et al. (2014) [14], Klowss et al. (2022) only modeled the diffusion inside the spheroid given that the nutrient concentration at the boundary is constant and equal to some maximum far-field concentration.

These diverse approaches demonstrate the existing inconsistency and ongoing challenge to accurately model the mass transport process in 3DCAs and motivates us to more precisely derive and characterize the amplification BC between 3DCAs and the surrounding media using theoretical and experimental methods. We use an effective diffusion model to describe the porous structure of 3DCAs and determine the corresponding diffusion coefficient and concentration. To quantify the amplification BC, we analytically demonstrate that the concentration of molecules inside the boundary is amplified relative to the concentration at the outer boundary by a factor that is greater than or equal to one, which we refer to as the *amplification factor*.

To provide an intuitive explanation for the boundary amplification factor, let us consider a 3DCA culture inside a microfluidic chip compartment or well that is exposed to a medium containing a known concentration of molecules. Due to the concentration gradient, molecules begin to enter the cell culture. However, the smaller diffusion coefficient within the 3DCA causes molecules that are freely diffusing in the well to become (relatively) trapped and absorbed within the 3DCA extracellular space. As a result, we expect that molecules accumulate inside boundary of the 3DCA, leading to a higher concentration than that outside the boundary. We want to emphasize that the amplification factor is a biophysical property related to diffusion, distinct from the (bio)chemical concepts of partition coefficient and solubility [16]. The partition coefficient is a measure of the ratio of concentrations of a compound in a mixture of two immiscible solvents at equilibrium, and typically serves to quantify the degree to which a chemical substance exhibits hydrophilicity ("water-loving") or hydrophobicity ("water-fearing"). Solubility measures the ability of substances to interact and form solutions.

We have provided a particle-based simulator (PBS) in which the Brownian motion of molecules is tracked and updated independently on both sides of the boundary, and passage across the boundary is treated accordingly. Following our preliminary results in [17], the PBS confirms the analytical result and reveals that a non-unitary amplification factor could lead to a noticeable impact on the molecule concentration in the medium when we add low-porosity cell cultures to the medium. When a 3DCA containing a lower concentration of target molecules is placed in a medium with a higher concentration of these molecules, our PBS reveals a rapid decrease in the concentration of the molecules in the culture medium. This rapid behaviour as a result of the amplification factor is a possible explanation for the observed initial offset in glucose concentration reported by Casas et al. (2022) [18] and similar 3DCA experiments. We leverage our proposed PBS to design a pilot experiment that could provide a

mechanistic explanation to the initial offset due to the amplification factor. Furthermore, this experimental method can be used to characterize the amplification factor.

For our experimental case study, we used liver spheroids as our 3DCAs and glucose as our target molecule. Prior to introducing the liver spheroids to a medium with a high glucose concentration of 11 mM, they were kept in two pre-culture media of volume 100 and 75 $\mu$L with a glucose concentration of 2.80 mM. Our experimental findings show a reduction in glucose concentration within the medium ($p = 0.008$ for 100 $\mu$L medium and $p = 0.06$ for 75 $\mu$L) over a very short time of 10 minutes. Additionally, our PBS results closely align with the experimental results, particularly for a media volume of 100 $\mu$L.

The remainder of this paper is organized as follows. We present the theoretical modeling including the mathematical proof. Then, we introduce the experimental methods employed, beginning with our approach to revealing the rapid drop in medium concentration, followed by the proposed PBS to confirm our theoretical results and the designed experiment. Finally, the results are discussed and the conclusions are presented.

## Theoretical methods

### Modeling of diffusion BCs at the border of 3DCA

A 3DCA with volume $V_s$ formed by $N_c$ cells including $N_c^1, \cdots, N_c^Q$ of $Q$ different types is considered. The culture's interior space is comprised of the cells and the extracellular space between the cells. Given that the volume of cell type $i$ is $V_c^i$, the total volume of the cells and the extracellular space inside the 3DCA are given by $\sum_{i=1}^{Q} V_c^i N_c^i$ and $V_s - \sum_{i=1}^{Q} V_c^i N_c^i$, respectively.

We model the 3DCA structure as a porous medium with volume $V_s$ whose porosity, $\epsilon$, is defined as the ratio of the extracellular space to the whole 3DCA volume, i.e.,

$$\epsilon = 1 - \frac{\sum_{i=1}^{Q} V_c^i N_c^i}{V_s}. \tag{1}$$

We assume that the cell culture is in a fluid medium that surrounds it and fills its extracellular space. The diffusive signaling molecules of type $A$ in the medium can diffuse into the extracellular space of the cell culture. In order to analyze the diffusion effect exclusively, we assume that there is no chemical reaction or binding occurring between the cells and target molecules within the cell culture structure. This assumption reasonably holds over a sufficiently short timescale when the culture is first exposed to the medium and the diffusion effect dominates. It is especially applicable when the rate of molecule binding and consumption is much slower than diffusion, such as in the case of glucose uptake by liver cells which is known to be much slower (i.e., hours) than the diffusion process in minutes [2].

Ideally, the porous culture structure acts as a net, enabling molecules outside the cell culture to pass through its border with a probability that depends on the surface porosity of the culture. This surface porosity can be estimated based on the geometric projection of volume porosity ($\epsilon$) in 3D onto 2D using $\epsilon_s = \epsilon^{\frac{2}{3}}$. Inside the 3DCA, the molecules diffuse via the curved paths of the extracellular space among the cells, which leads to a shorter net molecule displacement within a given time interval. Thus, macroscopic diffusion within the cell culture effectively differs from the diffusion within the free fluid outside it. Since the molecules traverse a shorter net path within the cell culture, the effective diffusion coefficient is smaller than the diffusion coefficient in the free fluid medium and molecules are more likely to be observed and sensed by the culture's cells (see Fig 1).

We assume that the extracellular space within the 3DCA is homogenized to model an effective diffusion environment throughout the entire culture volume. Given the diffusion

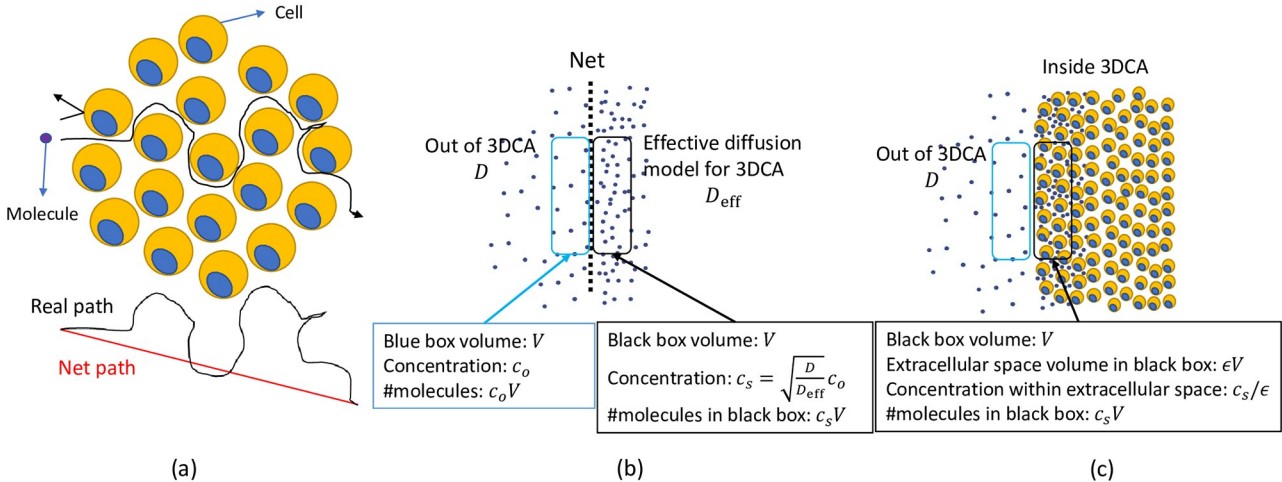

**Fig 1.** (a) The curved path of molecule trajectories within a spheroid leads to an effective diffusion coefficient smaller than the diffusion coefficient of the free medium, i.e., $D_{\text{eff}} < D$. (b) Boundary between media outside of 3DCA and the effective diffusion model for 3DCA which is a free medium diffusion environment with $D_{\text{diff}}$. The net (dashed line) at the boundary models the surface porosity and corresponding molecule reflection. The concentration and number of molecules at both sides are shown for a small volume very close to the boundary. (c) Boundary between media outside and inside the 3DCA that includes both the cells and extracellular space.

coefficient $D$ for molecules $A$ in the free fluid, the effective diffusion coefficient within the entire cell culture volume (homogenized environment) is given by [19]

$$D_{\text{eff}} = \frac{\epsilon}{\tau} D, \qquad (2)$$

where $\tau$ is the tortuosity, a measure of the transport properties of the porous medium, and is modeled as a function of cell culture porosity as $\tau = \frac{1}{\sqrt{\epsilon}}$ [20].

These two diffusion environments inside and outside the 3DCA are connected by defining two boundary conditions (BCs) at the border of the 3DCA. The first BC is the following flow continuity condition, which is applied at the border to ensure equal mass flux across the boundary:

$$D_{\text{eff}} \nabla c_s(\bar{r}, t) \cdot \hat{n} = D \nabla c_o(\bar{r}, t) \cdot \hat{n}, \qquad \bar{r} \in \partial\Omega, \qquad (3)$$

where $\partial\Omega$ denotes the boundary region of the cell culture, $\hat{n}$ is the normal vector at the border point $\bar{r}$, $c_o$ is the concentration of molecules outside the cell culture, and $c_s$ is the concentration of molecules inside the equivalent diffusion model of the cell culture, i.e., a free diffusion environment of volume $V_s$ with diffusion coefficient $D_{\text{eff}}$. Thus, the concentration inside the extracellular space within the cell culture is given by $c_s/\epsilon$. For the two diffusion environments, the second BC is characterized by the concentration ratio inside and outside of the 3DCA border as [7]

$$c_s(\bar{r}, t) = k c_o(\bar{r}, t), \qquad \bar{r} \in \partial\Omega, \qquad (4)$$

where $k$ has not been derived and is suggested to be determined experimentally [7]. This is a common BC model used for the border of a 3DCA with surrounding media. Despite its importance, this second BC for a 3DCA inside media has not been comprehensively or consistently characterized in the literature. However, the amplification factor should be derived as a function of medium porosity. It is important to highlight that while the amplification factor

characterizes the BC at the border, it influences the whole diffusion process both inside and outside the 3DCA by coupling the diffusion equations of both sides through the corresponding BC.

We refer to the BC (4) and $k$ as the *amplification BC* and *amplification factor*, respectively. The amplification BC accounts for the influence of the 3DCA on the diffusive transport of molecules and has a significant impact on the accuracy of the diffusion model. We previously used a PBS for a system with a spheroid inside an unbounded environment as two ideal diffusion environments with diffusion coefficients $D$ and $D_{\text{eff}}$ and, without any net barrier, the simulation results suggested $k = \sqrt{\frac{D}{D_{\text{eff}}}}$ [17]. Here, we theoretically prove that $k = \sqrt{\frac{D}{D_{\text{eff}}}}$ generally characterizes the amplification BC for a 3DCA inside a medium modeled as two diffusion environments separated by a border with porosity of $\epsilon_s$. Since the boundary is assumed to be ideal, a particle that passes through the border will follow its arrival direction and not change direction. As a result, we can assume a one-dimensional diffusion environment for the proof, without any loss of generality.

**Proposition 1**: For two ideal diffusion environments with diffusion coefficients $D$ and $D_{\text{eff}}$ separated by a surface with porosity $\epsilon_s$ and concentration functions $c_o(x), x \in [-\infty, 0]$ and $c_s(x), x \in [0, \infty]$, the amplification factor is equal to

$$k = \sqrt{\frac{D}{D_{\text{eff}}}}. \tag{5}$$

The proof is provided in the S1 Appendix.

Thus, for $k \neq 1$, a concentration discontinuity (i.e., jump) occurs at the boundary. Therefore, the concentration is amplified by factor $k$ when passing through the cell culture boundary. In the general scenario involving the interface between two arbitrary diffusion environments, it is possible to encounter a situation where $k < 1$, which is inconsistent with the concept of an *amplification* factor. We employ the term "amplification" to emphasize the elevated concentration observed within a 3DCA.

We note that $kc_o$ is the inner boundary concentration within the *equivalent free diffusion* environment with diffusion coefficient $D_{\text{eff}}$. Therefore, the concentration within the extracellular space inside of the cell culture boundary would be $\frac{kc_o}{\epsilon}$.

For simplicity of the presentation, we proposed Proposition 1 under the assumption of homogenous environments on both sides of the boundary. However, the proof holds for non-homogenous environments where the diffusion coefficient is a function of location. In such an environment, the diffusion coefficient variability across space can be managed by adjusting the time interval $\Delta t$, thus controlling the proximity to the border where the diffusion coefficient can be reasonably approximated as fixed.

## Rapid concentration reduction in culture medium

To provide an intuitive explanation for the boundary amplification factor, let us consider a 3DCA culture inside a microfluidic chip compartment or experimental well that is exposed to a medium containing a known concentration of molecules. Due to the concentration gradient, molecules begin to enter the cell culture. However, the smaller diffusion coefficient within the 3DCA causes molecules that are freely diffusing in the well to become (relatively) trapped and absorbed within the 3DCA extracellular space. In other words, due to the small effective diffusion coefficient inside the cell culture, it takes a longer time for molecules to exit.

As a result, we expect that molecules accumulate within the cell culture, thus reducing the concentration of the molecules in the medium, provided that there are no reactions between

cells and molecules that occur faster than the transient diffusion behavior. This phenomenon is a consequence of the amplification factor that comes from the smaller effective diffusion coefficient inside the cell culture. Obviously, the concentration reduction would be more significant for larger cell culture volumes and higher amplification factors. To provide a preliminary estimate of the concentration change, let us make some simplifying assumptions and perform some quick insightful calculations.

Let us consider a well that is filled with a medium of volume $V_m$ and a molecule concentration of $C_m$. Next, we introduce $N_s$ 3DCAs, such as spheroids, to the medium. Prior to this, the cell cultures were maintained in a pre-culture medium with a concentration $C_p < C_m$. Initially, the molecule concentration in the spheroids is expected to be $C_s = kC_p$, assuming no reaction or consumption of the molecules, with a homogeneous distribution of molecules within the spheroids.

We can assume that each spheroid contains $C_s V_s$ molecules at the time of being added to the well. After a brief period of time, numerous molecules are expected to have diffused into the spheroids due to the concentration gradient. We can further assume that molecule reactions with the cells are significantly slower than the diffusion rate within the spheroids. As a result, no molecule is lost due to reaction during the transient period when there is net diffusion of molecules into the spheroid.

Under these conditions, the molecule concentration within the spheroid will increase and eventually reach a local time equilibrium $C_s'$. At the same time, the molecule concentration in the medium decreases to a constant average level denoted by $C_m'$. We note that the simplifying assumptions in this subsection illustrate calculations where a concentration change outside the spheroid is readily measurable. We use these to design and justify our experiments and results. However, it is essential to emphasize that the amplification factor remains constant and influences the diffusion process, regardless of whether reactions within the spheroid occur at faster or slower rates. By local time equilibrium, we mean that the concentration throughout the whole spheroid(s) will be uniform for that short time. Given the boundary condition in (4), the concentration inside the spheroid will be $C_s' = kC_m'$. To obtain $C_s'$ and $C_m'$, we consider the molecule conservation in this closed system. The total number of molecules at time $t = 0$ was $C_m V_m + N_s C_s V_s$, which is equal to the number of molecules shortly thereafter at the local time equilibrium. Therefore, we have

$$C_m' V_m + N_s C_s' V_s = C_m V_m + N_s C_s V_s. \qquad (6)$$

By applying $C_s' = kC_m'$ and $C_s = kC_p$, we obtain

$$C_m' = \frac{C_m V_m + N_s k C_p V_s}{V_m + k N_s V_s}. \qquad (7)$$

Equivalently, the concentration reduction in the medium is obtained as

$$C_m - C_m' = \frac{\left(C_m - C_p\right) k N_s V_s}{V_m + k N_s V_s}. \qquad (8)$$

From (8), a higher $k$ value could further reduce the medium concentration inside the well while increasing the concentration inside the cell culture. However, this analysis is valid at equilibrium, and the exact time-point when this equilibrium occurs is unknown to us. On the other hand, if we wait too long to measure, then the biochemical reactions in the cell pathways may significantly affect the molecule concentration inside the spheroids. In particular, for our pilot experiment, the liver cells take up glucose molecules. To better clarify these points, we use

a PBS that is able to track transient molecule concentration and we use liver cells taking up glucose molecules in our experiment, which is known to be much slower (i.e., hours) than the diffusion process [2].

## Experimental methods

In the previous section, we revealed and analyzed rapid concentration reduction in culture medium property that can influence the outcomes and interpretations of experiments involving 3DCAs. To measure and characterize the amplification factor, we utilize this property to design an *in vitro* experiment in which the liver spheroids are exposed to a fluid medium with glucose molecules. We use a PBS that accurately emulates the experimental activity and enables us to gain further insights into the relevant parameters required to carry out the experiment.

### Particle-based simulator

We use liver spheroids created by differentiated HepaRG cells and human hepatic stellate cells following the method described by Bauer et al. (2017) [2] for our simulations and pilot experiment. S1 Fig displays images of six spheroids that were formed using this method, shown at two different scales. Each image displays the maximum projected area of a spheroid in 2D. To determine the spheroid's area within the image, the software tool ImageJ [21] was utilized, and the radius of a circle with equivalent area is used as the spheroid's radius. The average spheroid radius across these 6 samples was found to be $R_s = 226 \, \mu$m.

The total number of cells in each spheroid is assumed to be $N_c = 25000$ including 24000 HepaRG cells and 1000 human hepatic stellate cells, as reported by Bauer et al. (2017) [2]. The diameter of differentiated HepaRG cells has been reported as 17 $\mu$m [22]. The volume of HepaRG cells is calculated to be approximately $1.7 \times 10^{-15}$ m 3, as a mean reference of volumes assuming a spherical or cubic shape with this diameter, $2.5 \times 10^{-15}$ m 3 and $0.9 \times 10^{-15}$ m 3, respectively. Since the number of hepatic stellate cells in the spheroid is negligible in comparison to HepaRG cells, we can make a rough assumption that the volume of a hepatic stellate cell is approximately the same as a HepaRG cell, as this simplification does not significantly impact our calculations. Thereby, using $N_c = 25000$, we obtain $\epsilon = 0.1$ from (1). Correspondingly, $D_{\text{eff}}$ is related to $D$ by $D_{\text{eff}} = 0.032D$ from (2). Therefore, given $D = 10^{-9}$ outside the spheroid, we obtain $D_{\text{eff}} = 4.2 \times 10^{-11}$ m 2/s.

For our simulation, we considered a flat-bottomed well with a radius of $R_w = 3.2$ mm and a height of 6.2 mm. Forty spherically shaped spheroids with radius $R_s = 226 \, \mu$m were randomly distributed over the bottom of the well. We chose to use medium volumes of $V_m = \{100, 75\}$ $\mu$L. To demonstrate the amplification factor and resulting concentration reduction in the culture medium, we chose a high glucose concentration of 11.12 mM (hyperglycemia) [2] in the incubation medium and a low glucose concentration of 2.80 mM in the pre-culture medium to keep the spheroids alive. The large difference between these two concentrations helps to highlight the effect of the amplification factor.

To simulate the concentrations of 11.12 mM and 2.80 mM glucose in the culture and pre-culture media, respectively, with particle-based simulations, we would require a very large number of molecules and that is not computationally feasible. However, since the molecules move independently in the PBS, we do not need to use the exact number of molecules corresponding to these concentrations. Instead, we randomly placed $N_m = C_m V_m = 10^6$ molecules within the medium space and $C_s V_s = k C_p V_s$ molecules within each spheroid where $C_p = \frac{N_m}{V_m} \times \frac{2.80}{11.12}$, which leads to the same ratio of 11.12:2.80 between the medium and pre-culture medium. We normalized the concentration values by the initial concentration inside the medium, which was set to 1, in order to demonstrate the results.

In the PBS, time is divided into time steps of $\Delta t = 0.1$ s. In each time step, the molecule locations are updated following random Brownian motion. The molecules move independently in the 3-dimensional space, either in the medium or within the spheroids where the displacement of a molecule in $\Delta t$ s is modeled using Gaussian random variables both with zero mean and variances $2D\Delta t$ and $2D_{\text{eff}}\Delta t$, respectively, along each dimension (Cartesian coordinates).

In reality, the movement of molecules outside the spheroid may be affected by the porous spheroid surface. Molecules may pass through the extracellular spaces of the surface or reflect back if they hit other parts of the surface. For the simulation, the surface porosity of the spheroid is represented by the probability $\epsilon_s$, which is the likelihood that a hitting molecule will enter the surface. For molecules inside the spheroid, we assume an equivalent diffusion environment with an effective diffusion coefficient $D_{\text{eff}}$. It is noteworthy that the diffusion coefficient within a spheroid may vary according to the local cell size and density (e.g., it may be lower in a necrotic core with dead and dying cells). However, using the simplified homogenous model in PBS is justified due to the following reasons:

(i)The PBS uses an average effective diffusion coefficient for inside the 3DCA that mitigates the impact of variation. (ii), We focus on the initial 10 minutes when the molecules may not get very far from the boundary in the real experiment, particularly considering the smaller diffusion coefficient compared to the outside. The region of the spheroid near the surface is more likely to be uniform and have a comparable diffusion coefficient. (iii) Further, the protocol used to form the liver spheroids is based on [2] where the authors demonstrate the viability of the cells at the core. Thus, we expect that there is no necrotic region present.

We note that in this simplified environment model, the molecules can move freely inside the spheroid, and the porous structure effect is taken into account by using the effective diffusion coefficient. However, like the molecules outside, the molecules diffusing inside the equivalent environment may collide with the spheroid wall and exit with a probability of $\epsilon_s$. We note that the opening sites over the boundary enables the passage of molecules in both directions, i.e., from the outside to the inside of the spheroid and vice versa. Consequently, we assume an equal probability of $\epsilon_s$ for movement in either direction.

Considering the mismatch between the diffusion coefficients, we need to update the displacement vector of a molecule that crosses the spheroid boundary. For example, consider that a molecule in the medium outside a spheroid and its displacement vector during $\Delta t$ s is $(\Delta x, \Delta y, \Delta z)$ with a length of $d_T$ that would move the molecule into the spheroid. This vector has two parts: one part of length $d_o$ outside the spheroid and one part of length $d_i$ inside the spheroid. For the step inside the spheroid, the molecule diffuses at a slower pace, represented by a reduced diffusion coefficient in the model, which diminishes the size of the step by the square root of the ratio of diffusion coefficients. Then, the vector length for the inside part needs to be scaled by the factor $\sqrt{\frac{D}{D_{\text{eff}}}}$. As a result, the displacement vector is updated as follows

$$\frac{d_o + \left( \sqrt{\frac{D_{\text{eff}}}{D}} d_i \right)}{d_T} (\Delta x, \Delta y, \Delta z). \tag{9}$$

Similarly, if a molecule moves from inside to outside of the spheroid, then we need to update displacement vector outside according to

$$\frac{d_i + \left(\sqrt{\dfrac{D}{D_{\text{eff}}}}d_o\right)}{d_T}(\Delta x, \Delta y, \Delta z). \tag{10}$$

We also consider the cases where molecules diffusing within the well environment may hit the well walls or the top surface of the medium. In all these cases, the molecules are reflected back inside the well.

Fig 2(a) presents the normalized glucose concentration (NGC) both inside and outside the spheroid for medium volumes of 75 and 100 $\mu$L. The arrows indicate the ratio of NGC inside the spheroid to NGC outside, which is found to be approximately $k = \sqrt{\frac{D}{D_{\text{eff}}}}$, as derived in (5). In Fig 2(b), we show a higher resolution view of NGC in the medium for both volumes.

The concentration dynamic in the first 10 minutes (which is significantly shorter than the duration of the glucose uptake process by the spheroids) suggests that measuring the time-averaged NGC provides a good estimate of the NGC reduction due to the spheroids. Fig 2(c) shows that the predicted time-averaged GC for medium volumes of 75 and 100 $\mu$L over a period of 10 minutes is found to be 10.37 and 10.58, respectively, corresponding to a 5.7 and 3.8 percent GC reduction in the media, respectively. Therefore, in the *in vitro* experiment we took samples at 1, 5, and 10 minutes (min) after addition of spheroids to high-glucose medium to estimate the time-averaged NGC during the first 10 min, as chemical reactions may have a significant impact at longer timescales. We calculated the number of replicates required to achieve a power of 80 percent to distinguish these GC reductions from 11.12 mM, considering a variance of 0.15 for measurements and a significance level of 0.1. The required number of replicates was 3 and 4 for the two cases, respectively, to achieve a power of 80 percent. However, we conducted 8 replicates for each case to achieve a higher level of confidence. We note that the glucose concentration could potentially be measured in the cells but this would be an endpoint assay, i.e. the cells have to be lysed to release the glucose, hence it would require more cellular material. However, the experiments provided measure the concentration in the culture media surrounding the spheroids to infer the amplification factor.

## Liver spheroid formation and glucose assay

Formation of liver spheroids is based on the method published by Bauer et al. (2017) [2]. Differentiated HepaRGs (Lot HPR116239) were obtained from Biopredic International (Rennes, France). Primary human hepatic stellate cells (HHSteC), lot PFP, were purchased from BioIVT (Brussels, Belgium).

The differentiated HepaRGs were thawed and seeded confluently three days before spheroid formation. Standard HepaRG culture medium consisted of William's Medium E without glucose (PAN-Biotech, Aidenbach, Germany) supplemented with 10% foetal bovine serum (Gibco, ThermoFisher Scientific, Waltham, MA, USA), 11.12 mM glucose (Sigma–Aldrich, St. Louis, MO, USA), 5 $\mu$g/mL human insulin (Gibco), 2 mM L-glutamine (Corning), 50 $\mu$M hydrocortisone hemisuccinate (Sigma–Aldrich), 50 $\mu$g/mL gentamycin sulfate (Gibco) and 0.25 $\mu$g/mL Amphotericin B (Gibco). On the following day, the medium was renewed with HepaRG medium containing 2% dimethyl sulfoxide. The cells were maintained in this medium for two days until spheroid formation. HHSteC were expanded in Stellate Cell Medium, provided by ScienCell (Carlsbad, CA, USA). Cells were used in passage 3. Pre-culture was started two days before spheroid formation.

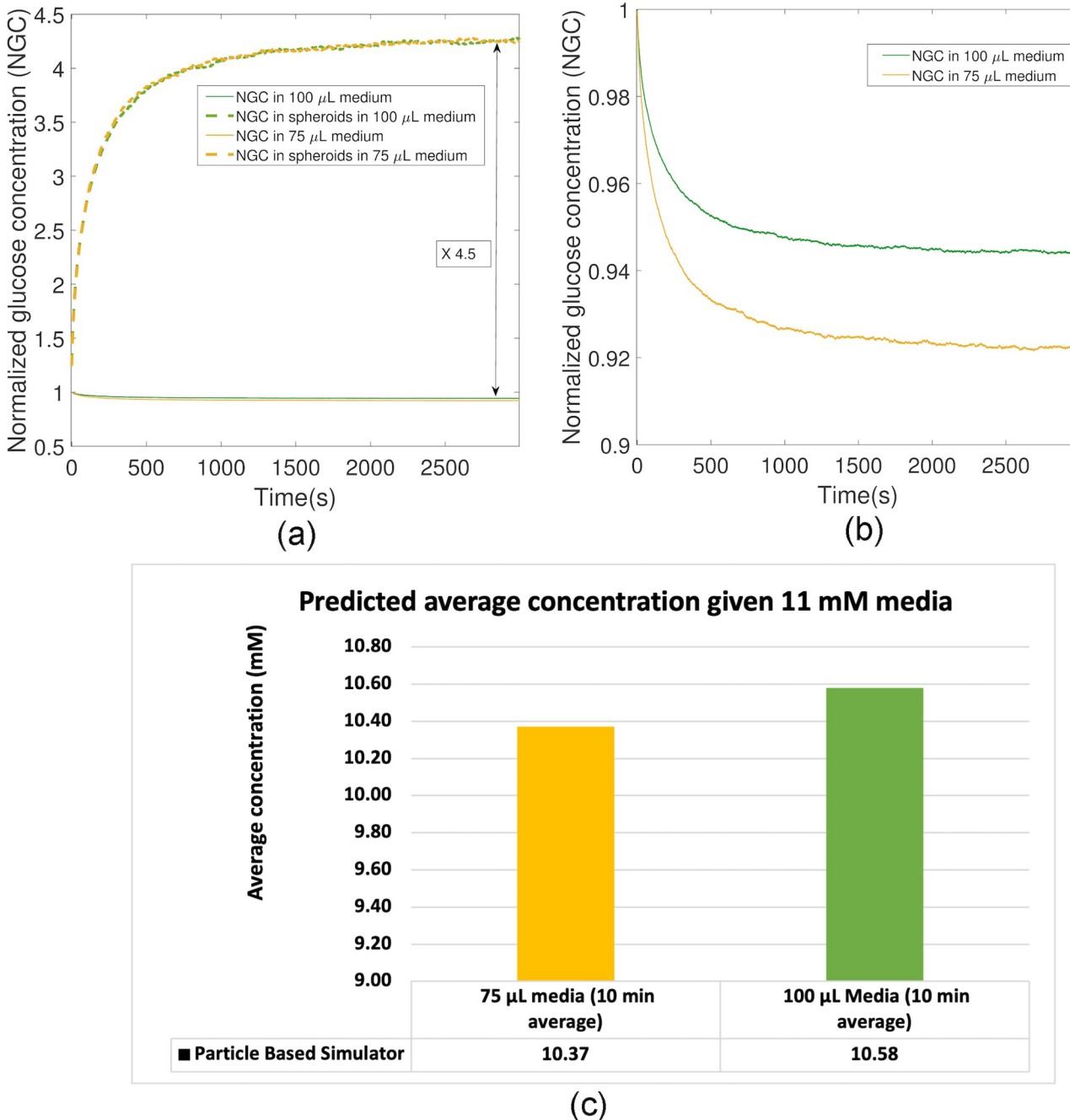

**Fig 2.** (a) Normalized glucose concentration (NGC) inside and outside the spheroid as a function of time for both medium volumes of 75 and 100 $\mu$L obtained by PBS. NGC inside the spheroids is the average over all 40 spheroids in the well. The ratio of NGC inside the spheroid to NGC outside is approximately 4.5. (b) A higher resolution view of NGCs in the medium for both volumes provided in Fig 2(a). (c) The predicted time-averaged GC for medium volumes of 75 and 100 $\mu$L after a period of 10 minutes based on the results in Fig 2(b).

Human liver spheroids were formed combining differentiated HepaRG cells and HHSteC using 384-well spheroid microplates (Corning, Lowell, MA, USA) in HepaRG medium. Briefly, 50 $\mu$L containing 24,000 hepatocytes and 1,000 HHSteC was pipetted into each well of the spheroid plate. The plate was centrifuged for 1 min at 300 xg and incubated at 37 °C and

5% CO2. Two days after seeding, 20 $\mu$L medium was removed and 50 $\mu$L fresh medium was added to the spheroids and five days after seeding 50% medium was renewed.

Six days after seeding, 40 spheroids were pooled into a 24-well ultra low attachement plate (Corning). The spheroids were washed once with 500 $\mu$L HepaRG medium with 2.80 mM glucose and 1 nM insulin. After the wash, 800 $\mu$L of this medium was added and spheroids were incubated for 2 h. Spheroids were then transferred to a 96-well flat bottom ultra low attachment plate (Corning) and medium was changed to 75 or 100 $\mu$L (8 replicates per volume) HepaRG medium with 11.12 mM glucose and 870 nM insulin. Taking Equation (8) into account, the large difference between the glucose concentrations of 2.80 mM in the pre-culture and 11.12 mM in the culture media is anticipated to result in a sharper change in glucose concentration within the culture media following the addition of 11.12 mM medium to the spheroids. Medium samples were taken after 1, 5, and 10 min and HepaRG medium with 11.12 mM glucose and 870 nM insulin was sampled as 0 min control. For samples where starting volume was 100 $\mu$L, a medium sample was also taken after 4 h. For samples with starting volume of 75 $\mu$L, all medium was removed after 10 min sampling and 100 $\mu$L new medium was added. From these incubations, the medium was sampled after 19 h. Fig 3 represents the schematic of the experiment.

After sampling, the medium was frozen and kept at -80 ˚C until analysis. Glucose concentrations in the samples were measured using Stanbio Glucose LiquiColor test. All samples were analysed undiluted according to manufacturer's instructions. In short, 5 $\mu$L sample or standard was added to a clear flat bottom 96-well plate (Nunc). The glucose reagent was pre-heated to 37 ˚C and the 95 $\mu$L reagent was added to start the reaction. The plate was centrifugated in short-spin to remove air bubbles and then incubated at 37 ˚C for 5 min. After incubation absorbance was directly measured at 520 nm on a Spectramax Plus reader and sample concentrations were calculated from the standard curve.

## Results

Fig 4 demonstrates the porosity ($\epsilon$) and amplification factor ($k$) as a function of the number of cells, $15000 < N_c < 25000$ in the spheroid. As observed in Fig 4, $k$ increases exponentially and $\epsilon$ demonstrates a linear decrease with an increase in $N_c$. For $N_c = 25000$, which is the approximate number of cells in HepaRG spheroids, we have $\epsilon = 0.12$ and correspondingly $k = 4.87$. This value of $k$ suggests a large concentration discontinuity at the spheroid boundary.

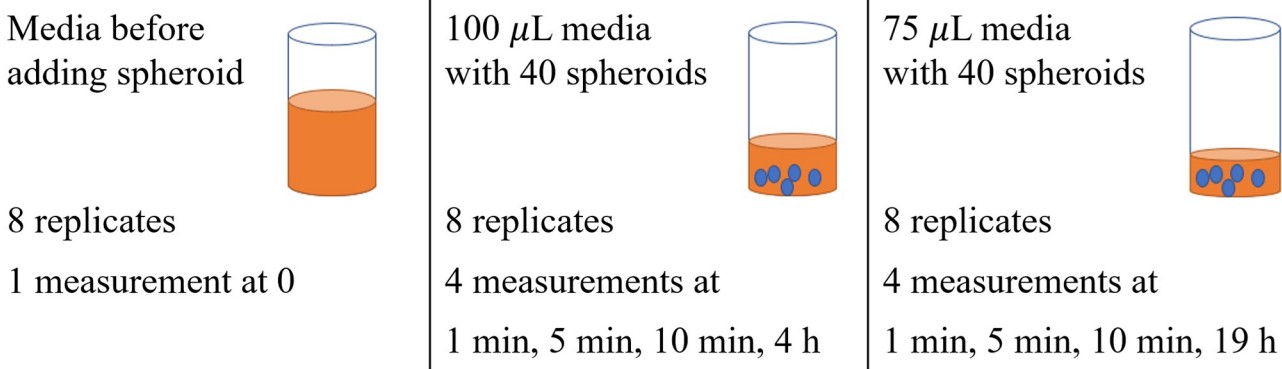

**Fig 3. Schematic representation of *in vitro* experiment.** Note that measurements taken at 4 h and 19 h are not considered in the evaluation of the amplification factor. These measurements are solely utilized to demonstrate the viability of the cells and the timescale of glucose uptake.

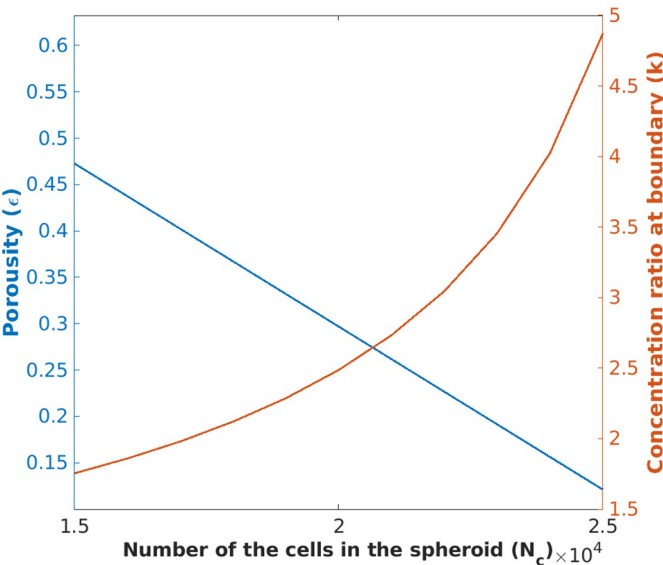

**Fig 4. Porosity ($\epsilon$) and concentration ratio at the boundary ($k$) versus the number of cells inside the spheroid ($N_c$).**

Fig 5 visualize the data used to generate the results depicted in Figs 6 and 7. Fig 5(a) and 5(c) illustrate the outcomes of 8 replicates with incubation volumes of 100 $\mu$L and 75 $\mu$L. For each replicate, the glucose concentration was measured at 1, 5, and 10 min after addition of spheroids to the high-glucose medium (the raw data is provided in S2 and S3 Tables). The concentration dynamics observed in the initial 10 minutes of the PBS results indicated that utilizing the time-averaged NGC provides a good estimate of the NGC reduction due to the spheroids. Additionally, this time-averaging approach helps to reduce the influence of the high data variability.

Fig 5(b) and 5(d) present the average of three time points at 1, 5, and 10 min (shown by green and orange bars, respectively) for each replicate and the statistical average (mean) of all replicates (shown by the purple line). The obtained average can be interpreted as the 10 min approximated average of glucose concentration. The mean glucose concentration for the 100 and 75 $\mu$L incubations is observed to be 10.67 and 10.86 M, respectively.

Fig 6(a) shows that the mean glucose concentrations (calculated based on measurements at 1, 5, and 10 min) in the medium before addition of spheroids, 100 $\mu$L, and 75 $\mu$L incubations are 11.12, 10.67, and 10.86, respectively. A box plot of these data is presented in Fig 6(b), which confirms that there are no outliers based on the "median and quartiles" method. The viability of the spheroids is evident from the reduction in glucose levels observed after 4 hours (for 100 $\mu$L) and 19 hours (for 75 $\mu$L) (provided in S1 Table and S2 Fig). Furthermore, the data shows a noticeable decrease in glucose concentration during the initial 10 minutes in both 75 and 100 $\mu$L incubations, which is more pronounced than the reduction observed after 4 hours due to glucose utilization by HepaRG cells. This finding supports the proposed theory of an amplification factor that leads to rapid glucose reduction over the timescale of initial transient diffusion. It also indicates that glucose absorption by HepaRG cells is a slow process and may not have a significant impact over this initial period. To test this hypothesis, we conducted two hypothesis tests.

The first hypothesis test compares the mean glucose concentration in the spheroid-free medium ($\mu_0$) to the mean observed from the 100 $\mu$L experiment ($\mu_{100}$), i.e., $H_0$: $\mu_0 = \mu_{100}$ and

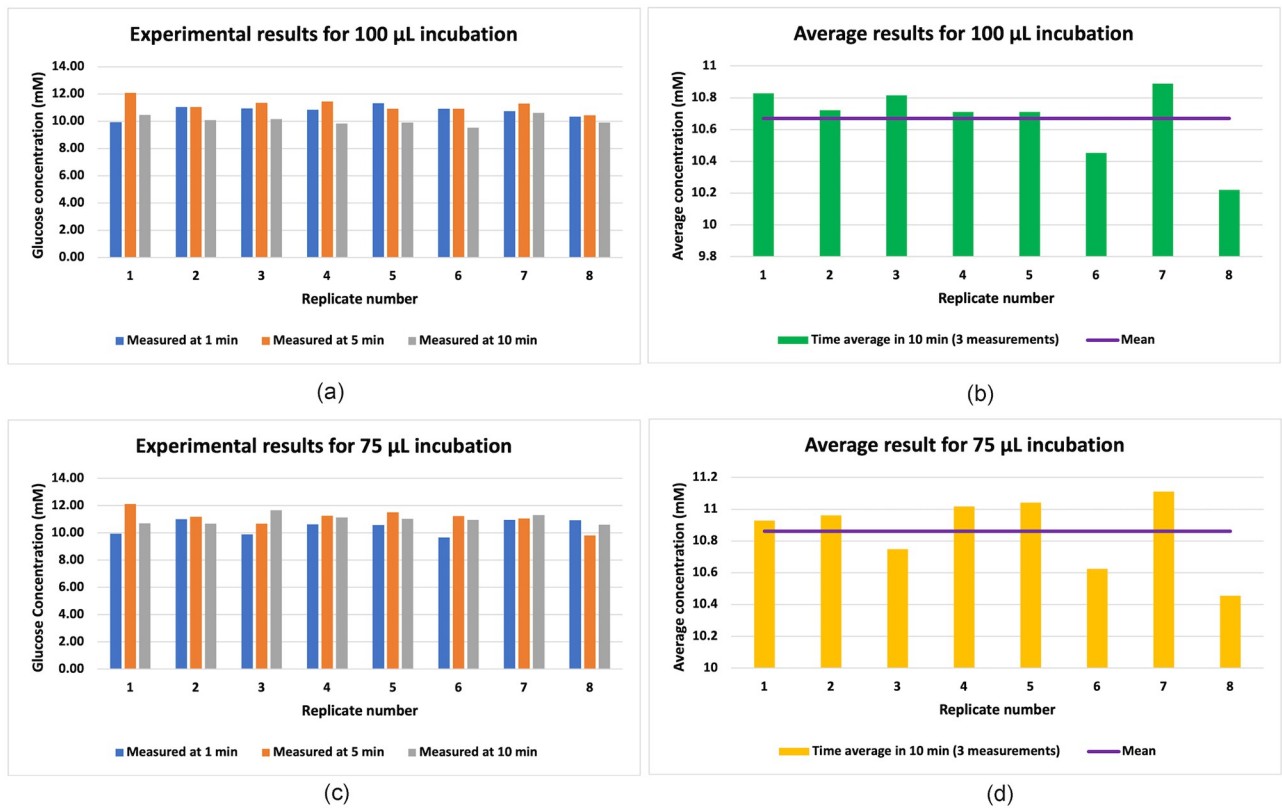

**Fig 5.** Glucose concentration measured at 1, 5, and 10 minutes after introducing the spheroid to the medium for 8 replicates of the 100 and 75 $\mu$L incubations, respectively (a,c). The 10-minute time-averages of measured concentration at 1, 5, and 10 min for each replicate and the statistical average (mean) of the replicates (shown by the purple line) (b,d). We note that a different scale has been used to display the average concentrations to better illustrate the variation.

$H_1: \mu_0 > \mu_{100}$. Similarly, the second hypothesis test compares the mean values for the 75 $\mu$L and spheroid-free experiment.

As mentioned, the mean glucose concentrations observed in the medium before addition of spheroids, 100 $\mu$L, and 75 $\mu$L incubations are 11.12, 10.67, and 10.86, respectively. The corresponding variances of the data samples are 0.15, 0.05, and 0.049. Due to the differing variances in the data sets, we use a two-sample t-test assuming unequal variances. We perform a one-tailed test due to the form of the alternative hypothesis.

The resulting p-values for the first and second tests are 0.008 and 0.06, respectively. These values indicate a significant difference for both hypotheses. Based on these results, we can conclude that the spheroids lead to a statistically significant reduction in glucose concentration during the initial 10 minutes.

In addition to the hypothesis testing, we have compared our experimental results with the glucose concentration predicted by the PBS in Fig 7. As observed, the PBS closely follows our experimental results for both cases.

## Conclusion

To enable the development of reliable biophysical models for 3DCAs, we utilized both theoretical and experimental methods to derive and characterize the amplification BC, which relates the concentration inside and outside the border between a 3DCA and its surrounding

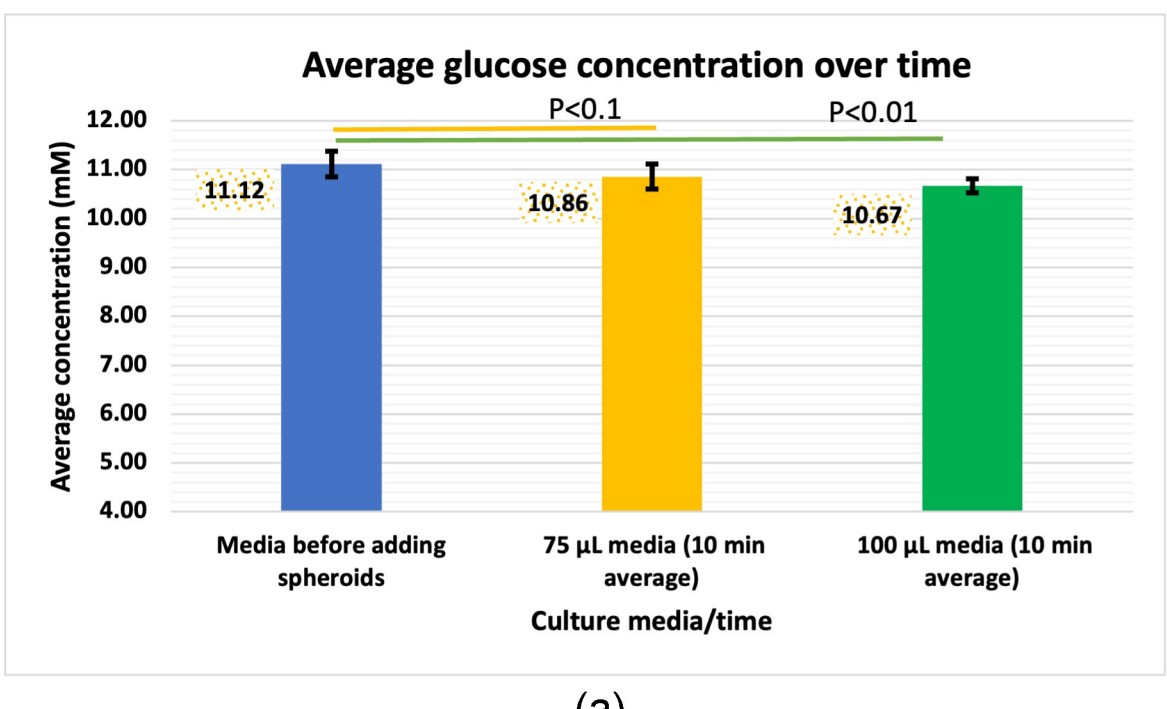

(a)

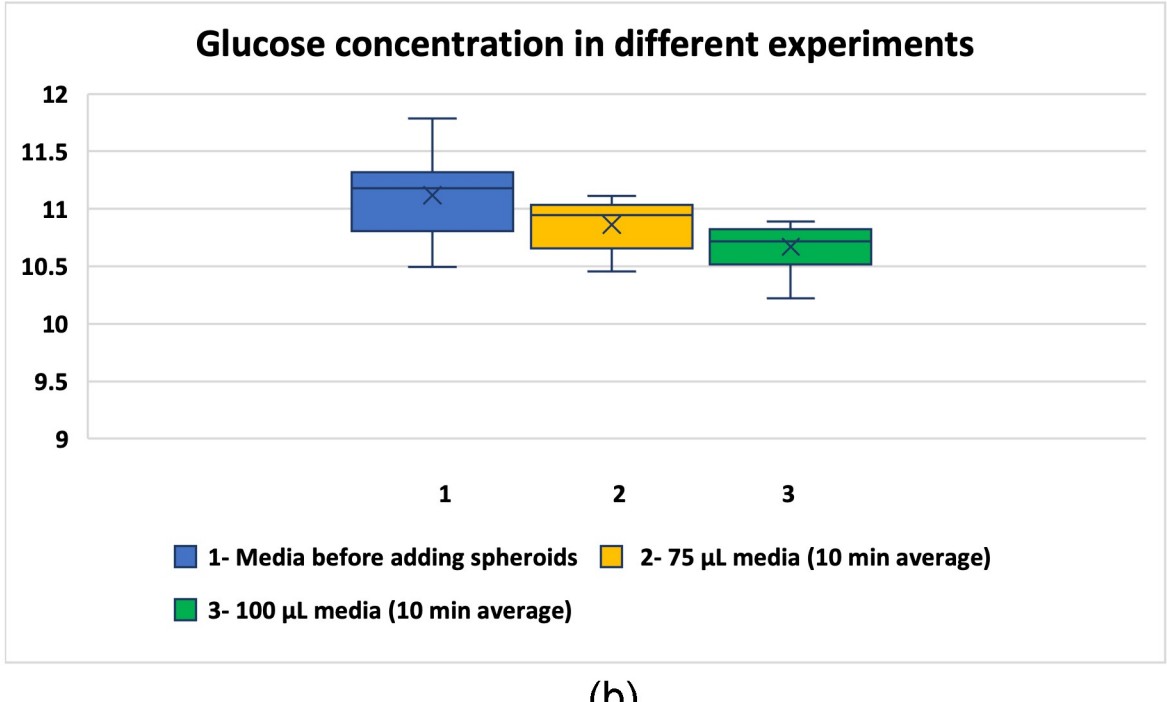

(b)

**Fig 6. Mean GC during 10 minutes after introducing the spheroid to the medium over 8 replicates of the 100 and 75 μL incubations.** (a) Mean GC in medium before adding spheroids and the mean 10 min time-averaged GC for the 100 and 75 μL incubations. (b) Boxplot of the results.

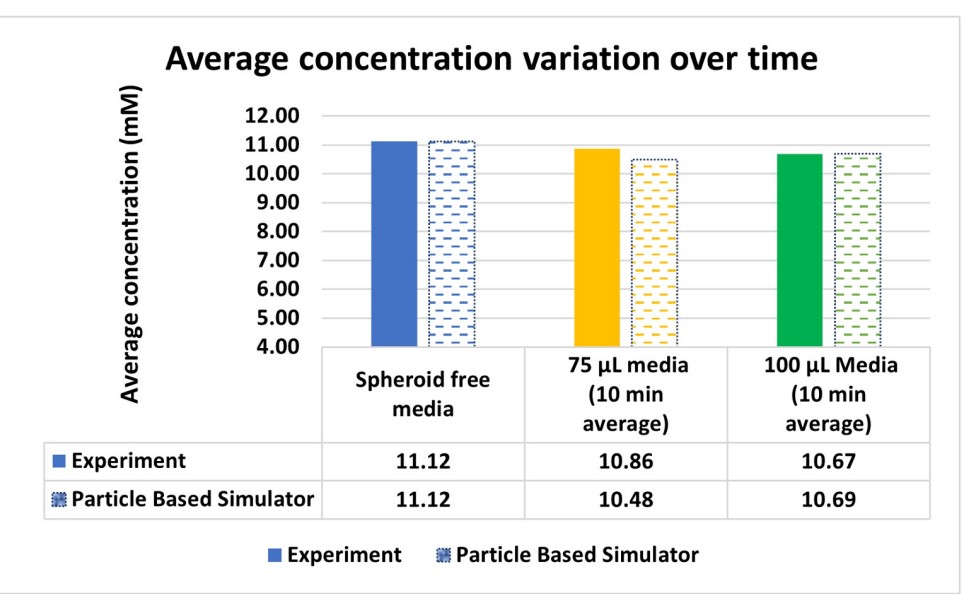

**Fig 7. Experimental glucose concentration in the media compared with the predicted values obtained from the proposed PBS, assuming an initial concentration of 11.12 mM.**

medium. Our biophysical theoretical analysis revealed a factor that characterizes the amplification BC and that this factor is a function of the two diffusion coefficients of the cell culture and medium. We confirmed this analytical result using a proposed PBS, which also showed a rapid decrease in the molecule concentration in the culture medium as a result of the amplification factor. To evaluate our approach, we conducted a pilot experiment using liver spheroids as the 3DCAs and glucose as the target molecule. Our study demonstrated a significant reduction in glucose concentration within the medium with $p = 0.008$ for 100 $\mu$L medium and $p = 0.06$ for 75 $\mu$L, in line with the PBS simulations.

The amplification factor revealed through our theoretical and experimental methods can have significant implications for biophysical models used in 3DCA experiments, including organ-on-a-chip models. Consideration of the amplification factor in such models would result in more accurate predictions of the biophysical models for 3DCAs, and consequently, aid in drug design and analysis of drug exposure within tissues. This factor may provide insight into the mechanisms behind tumor development and morphogenises. In particular, the packed structure of cancer tumors enables them to receive and trap a higher concentration of nutrients and oxygen molecules based on the amplification factor. Thus, this study could contribute to the development of novel approaches to manage and treat cancerous tissues. Furthermore, our study offers a generic experimental approach to quantify the amplification factor for different 3DCAs and contributes to a better understanding of this phenomenon. These types of advanced *in vitro* models will likely play a major role in future drug discovery, providing a human-cell based system that can reduce the number of animals used in research.

This was an initial pilot study using liver spheroids, and we require additional experimental data involving diverse cell types and varying conditions to more comprehensively characterize and capture the amplification property. The adoption of more precise measurement protocols and tools could prove invaluable in reducing the observed high variability.

## Supporting information

**S1 Table. Measured glucose concentration of medium before adding the spheroids.**
(ZIP)

**S2 Table. Measured glucose concentration at time points 1, 5, 10 min, and 4h in 100 $\mu$L incubations.**
(ZIP)

**S3 Table. Measured glucose concentration (GC) at times 1, 5, 10 min, and 19h in 75 $\mu$L incubations.**
(ZIP)

**S1 Fig.** (a) Microscopic images of three liver spheroids at 10X magnification (scale bar is 200 $\mu$m). (b) Microscopic images of three spheroids at 4X magnification (scale bar is 200 $\mu$m).
(ZIP)

**S2 Fig. Mean GC in medium before adding spheroids, mean 10 min time-averaged GC for the 100 and 75 $\mu$L incubations, mean GC at 4 hours (for 100 $\mu$L), mean GC at 19 hours (for 75 $\mu$L).**
(ZIP)

**S1 Appendix. Proof of Proposition 1.**
(ZIP)

**S2 Appendix. PBS code.**
(ZIP)

## Author Contributions

**Conceptualization:** Hamidreza Arjmandi, Kajsa P. Kanebratt, Liisa Vilén, Peter Gennemark, Adam Noel.

**Data curation:** Hamidreza Arjmandi, Kajsa P. Kanebratt.

**Formal analysis:** Hamidreza Arjmandi, Kajsa P. Kanebratt.

**Funding acquisition:** Adam Noel.

**Investigation:** Hamidreza Arjmandi, Kajsa P. Kanebratt.

**Methodology:** Hamidreza Arjmandi, Kajsa P. Kanebratt, Liisa Vilén, Peter Gennemark, Adam Noel.

**Project administration:** Liisa Vilén, Peter Gennemark, Adam Noel.

**Software:** Hamidreza Arjmandi.

**Supervision:** Liisa Vilén, Peter Gennemark, Adam Noel.

**Visualization:** Hamidreza Arjmandi, Kajsa P. Kanebratt.

**Writing – original draft:** Hamidreza Arjmandi, Kajsa P. Kanebratt.

**Writing – review & editing:** Hamidreza Arjmandi, Kajsa P. Kanebratt, Liisa Vilén, Peter Gennemark, Adam Noel.

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
