## [Decision Letter · Decision Letter 0]

27 Mar 2024

PONE-D-23-419393D Cell Aggregates Amplify Diffusion SignalsPLOS ONE

Dear Dr. Noel,

Thank you for submitting your manuscript to PLOS ONE. After careful consideration, we feel that it has merit but does not fully meet PLOS ONE’s publication criteria as it currently stands. Therefore, we invite you to submit a revised version of the manuscript that addresses the points raised during the review process.

We look forward to receiving your revised manuscript.

Kind regards,

Baeckkyoung Sung, Ph.D.

Academic Editor

PLOS ONE

Journal Requirements:

This work was supported by the Engineering and Physical Sciences Research Council [EP/V030493/1].

The authors acknowledge funding from the UK's Engineering and Physical Sciences Research Council, grant number EP/V030493/1. The funder had no role in study design, data collection and analysis, decision to publish, or preparation of the manuscript.

I have read the journal’s policy and the authors of this manuscript have the following competing interests: K.PK., L.V., P.G. are employees of AstraZeneca and hold stock/stock options.

We note that one or more of the authors are employed by a commercial company: AstraZeneca and hold stock/stock options. 

“The funder provided support in the form of salaries for authors, but did not have any additional role in the study design, data collection and analysis, decision to publish, or preparation of the manuscript. The specific roles of these authors are articulated in the ‘author contributions’ section.”

Additional Editor Comments:

The manuscript is recommended to be revised in accordance to the reviewers' comments. Particular improvements are expected in the aspects of modelling details and model validation.

Reviewers' comments:

Reviewer's Responses to Questions

**Comments to the Author**

1. Is the manuscript technically sound, and do the data support the conclusions?

Reviewer #1: No

Reviewer #2: Yes

Reviewer #3: Yes

Reviewer #4: Yes

2. Has the statistical analysis been performed appropriately and rigorously? 

Reviewer #1: No

Reviewer #2: Yes

Reviewer #3: Yes

Reviewer #4: Yes

3. Have the authors made all data underlying the findings in their manuscript fully available?

Reviewer #1: No

Reviewer #2: Yes

Reviewer #3: Yes

Reviewer #4: Yes

4. Is the manuscript presented in an intelligible fashion and written in standard English?

Reviewer #1: No

Reviewer #2: Yes

Reviewer #3: Yes

Reviewer #4: Yes

5. Review Comments to the Author

**Reviewer #1:** This paper is about diffusion signal amplification by 3D cell cultures. The topic tries to answer to the interesting question and this could draw attention from broad field of research. However the presented results are not convincing enough to support the author`s claims on the signal amplifiction.

This work summarizes preliminary results but to be published in any journal, I believe that more clear results must be provided. So I suggest that authors can make more measurements results and through statistical study.

The graphs must be provided with higher quality of image with errors bars as well.

I also suggest that authors should show microscopic photograph to show the cell cultures in 3D. They must also be linked to each measured results.

**Reviewer #2: **The present manuscript addresses a very important theme in biology, in which the scientist attempts to optimize human models for in vitro studies. 3D human cellular models are becoming the paradigm for drug screening and fundamental research since they are more realistic than animals and bypass the experimental animals’ ethics. Although a huge progress has been made in few years, as any emerging field in biology, there are several open questions and bottlenecks. The present manuscript focus in one of these, which is related to prediction models for signals diffusion and how to measure their accuracy.

The paper brings information on the development of a reliable biophysical model for the 3D cellular aggregates. More specifically on models for mass diffusion, which is critical for optimization and simulation of physiological aspects concerning drug discovery or even fundamental science.

The information provided on the manuscript is based in pilot experiments and it would be beneficial if authors could improve the sample number to consider a more reliable dataset. However, the data seems to be consistent and reproducible, so I do not have restrictions. Experimentally, the authors used a well stablish protocol for creating liver organoids and compare the results with simulated models. Also, they are one of few papers addressing the “3DCA border” in the models and making clear the importance in the mathematical model.

Overall the text is well written in good English, but I raise few minor points as suggestions:

- Although the introduction is very complete and may be beneficial for a non-expert reader, it should be shorted, referring to already published papers, especially in concerning the importance of 3DCA

- It should be clearer that authors are assuming a homogeneous extracellular space within the 3DCA. A conventional (2D) histology reveals it is not. But how it will impact the model is not clear for me and it deserves some discussion. Maybe consider a high-resolution X-ray tomography for 3D quantification of the inhomogeneity.

- In the “Results” section, there are missing some discussions of the presented data. For instance, in Fig. 4, 5 and 6.

- In line 426-432, the information of radius and volume was already presented in methodology and should be omitted.

- Please review lines 429-432. The information “… we have ϵ = 0.1 and 430 correspondingly k = 4.49…” is not presented in Figure 4, which shows the porosity and k for 25000 cells as below 0.1 and more than 5.5 , respectively.

- In the legend of Figure 6 should considering a first sentence as title.

- In Figure 6 consider including “glucose” in the plots’ titles (e.g. “Average glucose concentration…”)

- Lines 452-454 are almost identical to lines 447-449. Please correct by merging both information.

- Lines 465-468 brings introduction in statistical hypothesis test. It is not necessary and can be omitted. But, if author intend to keep, please move to methodology.

Considering the presented above and how complex and expensive is a 3D human cellular model, I think any improvement and optimization is beneficial. Then, my opinion is favorable for the publication of this manuscript.

**Reviewer #3:** Summary: In this paper, the authors explore the diffusive transport of glucose between 3D cell aggregates and the surrounding medium and focus on the influence of the boundary. Mathematically, the authors model diffusion in the medium as free diffusion, and diffusion in the 3D cell aggregate using free diffusion with an effective diffusion coefficient that assumes that transport in the 3D cell aggregate only occurs in the extracellular fluid. The authors also consider a discrete model of Brownian motion with different rates of movement for particles in the media and in the 3D cell aggregate. Experimentally, the authors incubate liver spheroids in a low glucose condition and then transfer them to a high glucose condition and measure the glucose concentration (focusing on the first 10 minutes). The authors then relate the results of the mathematical model to the experimental results. Overall, I feel that the study is of potential interest to the community. However, I feel that key methodological details are missing (particularly with the particle-based model) that would help in interpreting the results and that the structure of the manuscript should be modified. Further, I feel that it would help to clarify how K can be estimated from data and how the results can be extended to other situations. Additional detailed comments are included below.

Major comments:

[1] Key mathematical modelling details are not clear. In particular, it is not clear how to reproduce the particle-based simulator.

- Line 306 – How is k=4.49 obtained? It looks like the authors use the diffusivities from Line 330 and Eq 5. If so, this should be made clear at Line 306.

- Particle-based simulator. Do the authors model the full 3D experimental well or do they model one 3D cellular aggregate? If it is multiple 3D cellular aggregates, are they placed uniformly throughout the domain? Are the 3D cellular aggregates assumed to be spherical, or are other geometries chosen to represent their geometry? More details would be helpful.

[2] Structure of the manuscript. I am not sure that the heading “Experimental methods” accurately describes the content included in this section. “Rapid concentration reduction in culture medium” seems to be a mathematical modelling argument that would be better placed in the Theoretical methods section or the Results section. The particle-based simulator seems like it should be placed in the Theoretical methods section. Fig 2 and the associated discussion seems to better placed in the Results section.

[3] The mathematical proof in Appendix 1 would benefit from further justification on the points below.

- Particles located in x < -3sigma_0 have a non-zero probability of crossing the boundary. These are excluded from the argument. Similarly, for particles on the right-hand side.

- How is the probability determined? State which assumptions have been made. I think I know what these are from the description of the particle-based simulator, but this appendix should be self-contained for clarity.

- What is the definition of sigma_s?

[4] The literature review would benefit from examples of situations where one would expect jumps in the concentration of nutrient/oxygen/molecules at the media/3D cell aggregate interface and situations where one would not expect a jump in concentration. For example, it may be helpful to make comparisons to past experiments, e.g. Mueller-Klieser W, Freyer J, Sutherland R. Influence of glucose and oxygen supply conditions on the oxygenation of multicellular spheroids. British Journal of Cancer. 1986; 53:345–353.

[5] I feel that the conclusions (Lines 501-515) overstate the importance of this result.

-I agree that the result is of interest for 3D cellular aggregates surrounded by media over a very short duration. However, how do the results extend to typical experimental durations that range from hours to days where consumption is important? Further, how do the authors expect these results to extend to in vivo situations where 3D cellular aggregates are surrounded by other cells (where the difference in effective diffusivity would be smaller and k would more likely be approximately unity?)

- The amplification factor k appears to be calculated based on the diffusion coefficient in ref [2]. Line 511 – is the `quantification of the amplification factor’ referring to the result from Fig 2? In practice, how would one estimate K from experimental measurements that are obtained at a few time points? This seems to be an important explanation that is missing. Further, how could one assess the uncertainty associated with the estimate of K? The experimental measurements also appear to be measurements of the average concentration in the media or the spheroid. Presumably, this would impact estimates of K when the diffusive processes are not close to equilibrium?

[6] K is referred to as the amplification factor and experiments are performed transferring 3D cellular aggregates from a low glucose concentration to a high glucose concentration. What would happen if 3D cellular aggregates from a high glucose concentration to a low glucose concentrations? Is the term “amplification factor” still appropriate in this setting?

[7] Experimental methods - How is the glucose concentration measured in the spheroids? The methods appear to focus on how to measure the glucose concentration in the media.

[8] Results focus on early time measurements over the course of 10 minutes. Due to this short timescale are the authors able to provide more details on how the 3D cellular aggregates are transferred between the different glucose conditions. Does this impact the results?

Minor

[9] Line 87 – typo – “insider”

[10] Line 206-209 – it would be helpful to mention your previous study in the introduction.

[11] Line 258-259 – “Initially the molecule concentration in the spheroids is expected to be Cs < kCp” – Could you add further justification for why this is the case. This also appears on Line 276.

[12] Line 280 – Does equilibrium ever occur in this system?

[13] Line 300-301 – Please clarify the calculation used to obtain 1.7x10^-15.

[14] Line 353 – Please add extra justification for the scaling.

[15] Line 376 – Where did the variance of 0.15 come from? Does this agree with the experimental data?

[16] Line 395 – What is meant by “pre-culture”?

[17] Line 419 – What is meant by “standard”?

[18] Line 482 – “marginally significant” – The phrasing of these results in the conclusion on line 499-500 seems more appropriate.

[19] Fig 1 caption – is D_diff the same as D_eff?

[20] I am not sure that the author summary is required for PLoS One from the submission guidelines.

[21] Code availability – Is the code going to be made available?

**Reviewer #4: **This manuscript by Arjmandi et al presents a mixed computational/experimental investigation of glucose uptake in a 3D cell aggregate (3DCA) as an example of the more general mass transport case of nutrient transport within a 3DCA. The mathematical proof demonstrates that there is a locally increased concentration of glucose at the outer periphery of the 3DCA that corresponds to the mismatch between diffusion gradients in free media versus the 3DCA. This is an interesting and under-appreciated phenomenon in 3DCA-relevent studies, and this manuscript provides additional and useful information to this field. In general, the mathematical and theoretical considerations presented within this manuscript are technically sound and well-described. The authors then validate predictions of glucose uptake of 3DCA from media using serial timed media collections with differing initial media volume. The presented experimental conditions are well-described and technically sound. Predictions are then shown to match experimental conditions, validating the theoretical model of glucose uptake. In general, the manuscript is technically sound, scientifically rigorous, well-described, and well-written. However, there are a few additional points of consideration that the authors should address that would improve interpretations drawn from the manuscript.

1) There is extensive data supporting the amount of glucose taken up by the 3DCA by demonstrating loss of free glucose in the cell culture media. Timed-collection experiments show the decay in glucose that are attributable to transport within the 3DCA. These experiments validate predictions of the amount of glucose taken up by the 3DCA, but these studies do not validate the increased distribution of glucose concentration within the periphery of the 3DCA. As the major finding reported by this manuscript is the ‘amplification factor’ that characterizes this locally increased, it is curious that the authors did not perform any studies to validate this phenomenon. There are several experimental designs with labeled analogs of glucose (or dyes of similar molecular weight) that could have been utilized to validate this finding. The authors should either perform these types of validation studies or discuss this lack of validation to mitigate their conclusions.

2) The experimental conditions reflect an interesting thought experiment, but realistic cell culture conditions do not commonly apply such drastic step changes in glucose concentrations. In this manuscript, 3DCA are cultured in low glucose media and then transferred to high glucose media upon initiation of the experimental conditions. This was likely done to accentuate the initial glucose uptake in the 3DCA. However, it is difficult to draw comparisons with this experimental condition to that of a more common experimental condition in which 3DCA are maintained at similar glucose concentrations. How would this more common glucose concentration scenario be reflected? Will there still be the same ‘amplification factor’ if there is not a steep mismatch in glucose concentration? The authors need to discuss the interpretation of this data in a more common experimental scenario.

3) While not mentioned in the manuscript, it would be assumed that the local increase in glucose concentration at the periphery of the 3DCA would eventually equilibrate at steady state. For practical implications and application to future studies, how long is this ‘amplification’ maintained and what does the radial profile in glucose concentration look like prior to and during this equilibration period? This practical information is important to understand for applying this work to future 3DCA studies. If the time scale of equilibration of glucose concentration is much shorter than the time scale of the biological 3DCA culture experiment-of-interest, then this ‘amplification’ can be safely ignored. But if not, the findings in this paper are very important to consider. The authors should discuss time scale to better reflect the practical implication.

4) Minor: The font size in axes labels and legends in many of the figures is quite small and difficult to read.

6. PLOS authors have the option to publish the peer review history of their article (what does this mean?). If published, this will include your full peer review and any attached files.

Reviewer #1: No

Reviewer #2: No

Reviewer #3: No

Reviewer #4: No

---

## [Author Response · Author response to Decision Letter 0]

14 Jun 2024

The rebuttal letter responding to each point from the Reviewers is in the uploaded Response to Reviewers document.

---

## [Decision Letter · Decision Letter 1]

23 Jul 2024

PONE-D-23-41939R13D Cell Aggregates Amplify Diffusion SignalsPLOS ONE

Dear Dr. Noel,

Thank you for submitting your manuscript to PLOS ONE. After careful consideration, we feel that it has merit but does not fully meet PLOS ONE’s publication criteria as it currently stands. Therefore, we invite you to submit a revised version of the manuscript that addresses the points raised during the review process.

We look forward to receiving your revised manuscript.

Kind regards,

Baeckkyoung Sung, Ph.D.

Academic Editor

PLOS ONE

Journal Requirements:

Additional Editor Comments:

The authors are invited to an additional revision of the manuscript based on the reviewer's comments.

Reviewers' comments:

Reviewer's Responses to Questions

**Comments to the Author**

1. If the authors have adequately addressed your comments raised in a previous round of review and you feel that this manuscript is now acceptable for publication, you may indicate that here to bypass the “Comments to the Author” section, enter your conflict of interest statement in the “Confidential to Editor” section, and submit your "Accept" recommendation.

Reviewer #2: All comments have been addressed

Reviewer #3: (No Response)

2. Is the manuscript technically sound, and do the data support the conclusions?

Reviewer #2: Yes

Reviewer #3: Yes

3. Has the statistical analysis been performed appropriately and rigorously? 

Reviewer #2: Yes

Reviewer #3: Yes

4. Have the authors made all data underlying the findings in their manuscript fully available?

Reviewer #2: Yes

Reviewer #3: Yes

5. Is the manuscript presented in an intelligible fashion and written in standard English?

Reviewer #2: Yes

Reviewer #3: Yes

6. Review Comments to the Author

Reviewer #2: The current revised version (R1) of the manuscript addressed all issues I raised previously.

I can see a more detailed description of the methodology and discussion, overall the text.

Specifically:

- introduction is now more objective.

- data presented in Fig 4, 5 and 6 is now discussed

- minor issues were addressed

My opinion remains favorable to publication since there is a contribution to the field of organs-on-chip, improving the quality of the system, which let it be more reliable to physiological conditions.

Reviewer #3: The authors have thoughtfully and systematically addressed the comments that I raised. I have one comment outstanding that should be addressed prior to publication. I look forward to seeing the impact of this work in the field.

[1] Follow up to the authors response to Comment 14 regarding the scalings in Eq (9) and (10). The authors response states that for the step inside the spheroid the molecule diffuses at a slower pace which diminishes the step by the square root of the diffusion coefficients. In Eq (9), D > Deff so sqrt(D/Deff) > 1 will increase the step inside the spheroid rather than decrease it. Should the scaling be sqrt(Deff/D) instead here? Similarly, in Eq (10) should the scaling be sqrt(D/Deff) rather than sqrt(Deff/D)? If I am missing something additional clarification in the manuscript would be helpful. How does this influence the results?

7. PLOS authors have the option to publish the peer review history of their article (what does this mean?). If published, this will include your full peer review and any attached files.

Reviewer #2: No

Reviewer #3: No

---

## [Author Response · Author response to Decision Letter 1]

10 Aug 2024

The previous decision only flagged a single issue. As discussed in the Response to Reviewers, we have corrected only a textual error in (9) and (10).

---

## [Decision Letter · Decision Letter 2]

26 Aug 2024

3D Cell Aggregates Amplify Diffusion Signals

PONE-D-23-41939R2

Dear Dr. Noel,

We’re pleased to inform you that your manuscript has been judged scientifically suitable for publication and will be formally accepted for publication once it meets all outstanding technical requirements.

Kind regards,

Baeckkyoung Sung, Ph.D.

Academic Editor

PLOS ONE

Additional Editor Comments (optional):

The revised manuscript has well addressed the reviewer's comments.

Reviewers' comments:

Reviewer's Responses to Questions

**Comments to the Author**

1. If the authors have adequately addressed your comments raised in a previous round of review and you feel that this manuscript is now acceptable for publication, you may indicate that here to bypass the “Comments to the Author” section, enter your conflict of interest statement in the “Confidential to Editor” section, and submit your "Accept" recommendation.

Reviewer #2: All comments have been addressed

Reviewer #3: All comments have been addressed

2. Is the manuscript technically sound, and do the data support the conclusions?

Reviewer #2: Yes

Reviewer #3: Yes

3. Has the statistical analysis been performed appropriately and rigorously? 

Reviewer #2: Yes

Reviewer #3: Yes

4. Have the authors made all data underlying the findings in their manuscript fully available?

Reviewer #2: Yes

Reviewer #3: Yes

5. Is the manuscript presented in an intelligible fashion and written in standard English?

Reviewer #2: Yes

Reviewer #3: Yes

6. Review Comments to the Author

Reviewer #2: My previous recommendation was favorable to publish the manuscript.

This new version has just few minor modifications, addressed by authors.

I still recommend the publication.

Reviewer #3: The authors have addressed the comment that I raised, which turned out to be a typographical error with no impact on the results. I look forward to seeing the impact of this work in the field.

7. PLOS authors have the option to publish the peer review history of their article (what does this mean?). If published, this will include your full peer review and any attached files.

Reviewer #2: No

Reviewer #3: No

---

## [Editor Report · Acceptance letter]

2 Sep 2024

PONE-D-23-41939R2 

PLOS ONE

Dear Dr. Noel, 

I'm pleased to inform you that your manuscript has been deemed suitable for publication in PLOS ONE. Congratulations! Your manuscript is now being handed over to our production team.

Kind regards, 

on behalf of

Dr. Baeckkyoung Sung 

Academic Editor

PLOS ONE